# CTLA-4 antibody-drug conjugate reveals autologous destruction of B-lymphocytes associated with regulatory T cell impairment

Musleh M Muthana[1,2]*, Xuexiang Du[3], Mingyue Liu[1,2], Xu Wang[1,2], Wei Wu[4], Chunxia Ai[3], Lishan Su[1,2,5,6], Pan Zheng[4]*, Yang Liu[4]*

[1]Division of Immunotherapy, Institute of Human Virology, University of Maryland School of Medicine, Baltimore, United States; [2]Department of Pharmacology, University of Maryland School of Medicine, Baltimore, United States; [3]Key Laboratory of Infection and Immunity of Shandong Province & Department of Immunology, School of Basic Medical Sciences, Shandong University, Jinan, China; [4]OncoC4, Inc, Rockville, United States; [5]Division of Virology, Pathogenesis and Cancer, Institute of Human Virology, University of Maryland School of Medicine, Baltimore, United States; [6]Department of Microbiology & Immunology, University of Maryland School of Medicine, Baltimore, United States

*For correspondence:
MMuthana@ihv.umaryland.edu
(MMM);
Pzheng@oncoc4.com (PZ);
yangl@oncoc4.com (YL)

## Abstract

Germline CTLA-4 deficiency causes severe autoimmune diseases characterized by dysregulation of Foxp3[+] Tregs, hyper-activation of effector memory T cells, and variable forms auto-immune cytopenia including gradual loss of B cells. Cancer patients with severe immune-related adverse events (irAE) after receiving anti-CTLA-4/PD-1 combination immunotherapy also have markedly reduced peripheral B cells. The immunological basis for B cell loss remains unexplained. Here, we probe the decline of B cells in human CTLA-4 knock-in mice by using anti-human CTLA-4 antibody Ipilimumab conjugated to a drug payload emtansine (Anti-CTLA-4 ADC). The anti-CTLA-4 ADC-treated mice have T cell hyper-proliferation and their differentiation into effector cells which results in B cell depletion. B cell depletion is mediated by both CD4 and CD8 T cells and at least partially rescued by anti-TNF-alpha antibody. These data revealed an unexpected antagonism between T and B cells and the importance of regulatory T cells in preserving B cells.

## eLife assessment

This **valuable** study presents presents **solid** evidence that an anti-CTLA-4 antibody drug conjugate transiently depletes circulating B cells in a mouse model, showing how dysregulation of the T cell immune system can impact B cell homeostasis. The work will be of broad interest to immunologists and medical biologists, but a major limitation is that the mechanism of B-cell reduction remains unclear, as evidence of killing of B-cells by T-cells is not presented.

## Introduction

Foxp3 is a master regulator of regulatory T cells (Tregs) and its mutations result in fatal autoimmune disease in mice and human (*Bennett et al., 2001*; *Brunkow et al., 2001*; *Fontenot et al., 2003*; *Wildin et al., 2001*; *Wildin et al., 2002*). Among many known functions, Foxp3 is a transcriptional

factor for expression of CTLA-4, which is constitutively expressed in Tregs (*Tivol et al., 1995*; *Waterhouse et al., 1995*). *Ctla4* deletion in mice phenocopies that of Foxp3. In humans, CTLA-4 deficiency caused by autosomal heterozygous mutation of the CTLA-4 gene (*Kuehn et al., 2014*; *Schubert et al., 2014*) or mutations in recycling partner LPS-Responsive beige-like anchor (LRBA) protein (*Lo et al., 2015*) are associated with severe autoimmune diseases. Although CTLA-4 can be expressed at lower levels in other cell types and has been suggested as a negative regulator for naïve T cell activation, lineage-specific deletion of the *Ctla4* gene in Foxp3⁺ Tregs results in development of systemic lymphoproliferation, fatal autoimmune disease, and potent tumor immunity (*Wing et al., 2008*). These data are consistent with the notion that the predominant function of CTLA-4 is Treg-intrinsic.

A largely overlooked area is the cross-regulation between B cells and regulatory T cells. CTLA-4 conditional null mice and those with depletion of Tregs showed increase in germinal center B cells and heightened antibody responses with some reports showing a decrease B cell percentage (*Wing et al., 2014*; *Sage et al., 2014*; *Klocke et al., 2016*; *Leonardo et al., 2012*). B cell loss is a common feature in genetic mutations of Foxp3, scurfy mice, Foxp3 knockout mice or Treg depletion (*Clark et al., 1999*; *Chen et al., 2010*; *Chang et al., 2012*; *Riewaldt et al., 2012*; *Leonardo et al., 2010*; *Pierini et al., 2017*). Among cancer patients who received immunotherapy, early change of circulating B cells of patients who received combination immunotherapy anti-CTLA-4/PD-1 correlated to irAE (*Das et al., 2018*). The change in B cells included decline of circulating B cells, and increase in CD21 low B cells and plasmablasts (*Das et al., 2018*).

The correlation between B cell loss and immune activation associated with defective Foxp3-CTLA-4 function remains unexplained. Since both CTLA-4 and Foxp3 are largely expressed outside of B cell compartment, and since this pathway is the master regulatory of Treg function, we hypothesized that B cell loss maybe associated with activation of T cells that are autodestructive of B cells. This hypothesis is noteworthy as no autodestructive T cells for B cells have been described. To test this hypothesis, we generated anti-CTLA-4 ADC and showed that the ADC caused selective depletion of Tregs in mice and a marked reduction of B cells. Remarkably, activation of CD4 and CD8 T cells is the underlying cause of B cell depletion. Our data explains the B cell loss associated with Foxp3 and CTLA-4 dysfunction and suggested an unexpected antagonism between T cells and B cells.

## Results

### CTLA-4 antibody-drug conjugate depleted regulatory T cells and B cells

Anti-CTLA-4 antibody Ipilimumab (Ipi) or human IgGFc (hIgGFc) control were conjugated to a well-known drug payload DM1, emtansine, to furnish the corresponding ADCs: Ipilimumab-DM1 (Ipi-DM1) and hIgGFc-DM1 with corresponding size shifts in the SDS-gel (*Figure 1A*). The drug to antibody ratio (DAR) was calculated based on experimentally determined extinction coefficient A280 and A252 for each antibody and reported values for DM1 (*Supplementary file 1*). The DAR from various conjugation is 3.2 for Ipi-DM1 and 1.7 for hIgGFc-DM1 control (*Supplementary file 1*), as expected based on the sizes of the proteins. The binding for Ipi-DM1 ADC was evaluated by ELISA binding to immobilized His-hCTLA-4 and by flow cytometry using hCTLA-4 expressing CHO cells (CHO-hCTLA-4). Ipi-DM1 binding was found comparable to the parent antibody Ipi (*Figure 1B and C*). Specific killing of CTLA-4-expressing cells by Ipi-DM1 was assessed in human CTLA-4 expressing CHO (CHO-hCTLA-4) and wild-type CHO (CHO-WT) cells in vitro (*Figure 1—figure supplement 1*), which showed Ipi-DM1 reduced viability of CLTA-4 expressing CHO but not wild-type cells. As expected, there was no change in viability for either CHO cell lines when treated with Ipi (*Figure 1—figure supplement 1*).

In order to evaluate the in vivo effects of CTLA-4 ADC on Tregs, we treated human CTLA-4 knock-in (*Ctla4*ʰ/ʰ) mice with control hIgGFc or Ipi-DM1 ADC (*Figure 1D*). Peripheral blood was stained for flow cytometry and gated on CD45 and T-cell subset, including CD8, CD4, CD4 Foxp3⁺ and CD4 Foxp3⁻ subsets (*Figure 1—figure supplement 2A*). We found that Ipi-DM1 ADC significantly depleted Foxp3⁺ Tregs as indicated by the reduction in percentage and cell number of Foxp3⁺ CD4 T cells when compared to hIgGFc control (*Figure 1E*). In addition, total CTLA-4 levels and Ki67 staining in Tregs were decreased compared to control group (*Figure 1F and G*). In contrast, Ipi-DM1 ADC did not alter either % of CD4 Foxp3⁻ cells among CD45⁺ leukocytes or CTLA-4 levels on the subset, but did slightly decrease the cell number of CD4 Foxp3⁻ cells when compared hIgGFc control (*Figure 1—figure supplement 3A and B*). Unexpectedly, Ipi-DM1 ADC caused a dramatic decline of B cells as

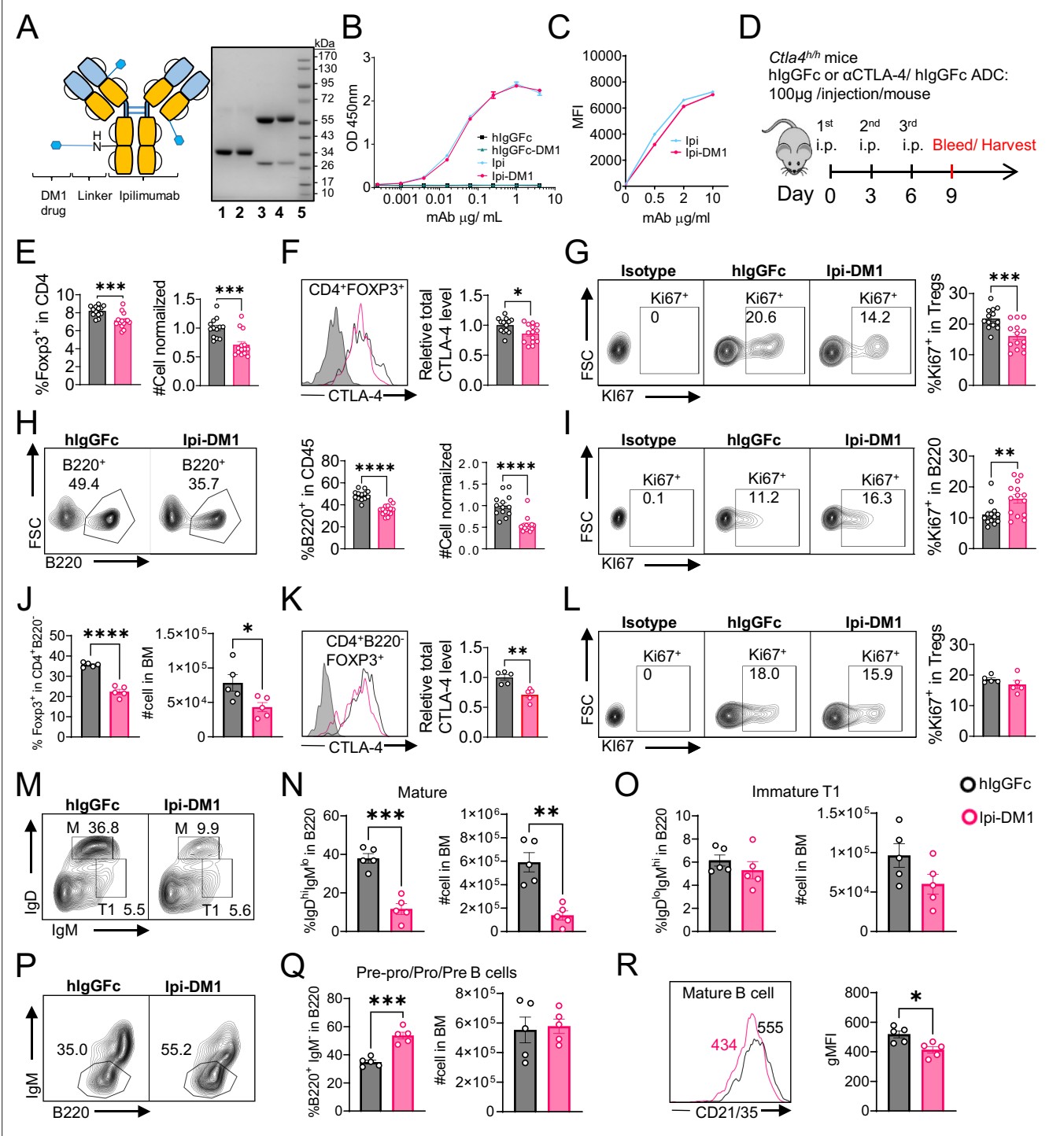

**Figure 1.** CTLA-4 antibody-drug conjugate impairs regulatory T-cell and leads to B-cell depletion. (**A**) SDS-Reducing gel showing size shift after DM1 conjugation. Lanes (left to right) are 1 hIgGFc, 2 hIgGFc-DM1 antibody-drug conjugate (ADC), 3 Ipilimumab (Ipi), 4 Ipilimumab-DM1 (Ipi-DM1) antibody-drug conjugate (ADC), and 5 Ladder. (**B**) ELISA binding of hIgGFc, hIgGFc-DM1, Ipi, and Ipi-DM1 to pre-coated 1 µg/mL His-hCTLA-4 and detected with anti-hIgG-HRP (n=2). (**C**) Flow cytometry binding of Ipi and Ipi-DM1 to hCTLA-4 expressing CHO cells (CHO-hCTLA-4) and detected with anti-hIgG-AF488 (n=1). (**D**) Diagram of experimental design, *Ctla4^h/h* mice were treated intraperitoneally (i.p.) (100 µg/mouse) with hIgGFc or Ipi-DM1 every 3 days and mice were bled or harvested for bone marrow extraction on day 9 for downstream flow analysis. (**E–I**) flow data analysis of day 9 peripheral blood, Tregs defined (CD45+ CD4+ Foxp3+) and B cells defined (CD45+ B220+). (**E**) % Foxp3+ in CD4 and normalized cell number. (**F**) Relative CTLA-4 Level in Tregs. (**G**) % Ki67+ in Tregs. (**H**) FACS profiles depicting gating strategy after gating on CD45+ and data summaries of % B220+ in CD45 and normalized cell number. (**I**) % Ki67+ in B cells. (**J–R**) Flow data analysis of day 9 bone marrow, Tregs defined (CD4+B220-Foxp3+) and B220+ B cell subtypes defined

*Figure 1 continued on next page*

*Figure 1 continued*

mature (**M**) (IgD$^{hi}$ IgM$^{low}$), immature transitional type 1 (**T1**) (IgD$^{low}$IgM$^+$), and Pre-pro/Pro/Pre B (B220$^+$IgM$^-$). (**J**) % Foxp3$^+$ in CD4$^+$B220$^-$ and absolute cell number. (**K**) Relative CTLA-4 Level in Tregs. (**L**) % Ki67$^+$ in Tregs. (**M**) FACS profile gating of M and T1 B cells, (**N**) % of mature B cells in B220 and absolute cell number summaries, (**O**) % of T1 immature B cells in B220 and absolute cell number summaries. (**P**) FACS profile gating of Pre-pro/Pro/Pre B cells. (**Q**) % of Pre-pro/Pro/Pre B cells in B220 and absolute cell number summaries. (**R**) CD21/35 expression in mature B cells in bone marrow. (**B, C**) Representative data of two or more repeats. (**E–I**) Data combined from two independent experiments (n=13–14). (**J–R**) Data representative of two independent experiments (n=5). Data analyzed using an unpaired two-tailed Student's t test and represented as mean ± SEM. *p<0.05, **p<0.01, ***p<0.001, ****p < 0.0001.

The online version of this article includes the following source data and figure supplement(s) for figure 1:

**Source data 1.**

**Figure supplement 1.** CTLA-4 antibody-drug conjugate killing is CTLA-4 specific.

**Figure supplement 2.** Gating strategy for T cells.

**Figure supplement 1—source data 1.**

**Figure supplement 3.** CD4-nonTregs and CTLA-4 expression.

**Figure supplement 3—source data 1.**

**Figure supplement 4.** B-cell depletion is not mediated by DM1 payload off-target release.

**Figure supplement 4—source data 1.**

**Figure supplement 5.** B-cells do not express CTLA-4.

**Figure supplement 6.** Tregs and B cells in spleen and lymph nodes.

**Figure supplement 6—source data 1.**

**Figure supplement 7.** IgM negative B-cell subtypes in bone marrow.

**Figure supplement 7—source data 1.**

percentages of CD45 and cell number in the periphery (*Figure 1H*). B cells from mice treated with Ipi-DM1 exhibited higher proliferation than those from hIgGFc treated control mice (*Figure 1I*).

Since a dramatic loss of B cells and a slight decrease in CD4-nonTregs in Ipi-DM1 treated mice were observed, we sought ensure that this phenomenon was not a result of downstream release of payload drug DM1. As shown in *Figure 1—figure supplement 4A*, hIgGFc-DM1 ADC (payload control) did not alter Treg percentage and cell number. A slightly higher level of CTLA-4 was noted in CD4 subset, while Treg proliferation did not change significantly compared to control (*Figure 1—figure supplement 4B and C*). No other effects on T cell subsets were noted (*Figure 1—figure supplement 4D and E*). Importantly, B cell number and proliferation were unaffected by hIgGFc-DM1 ADC treatment (*Figure 1—figure supplement 4F and G*). Additionally, to ensure B cell depletion was not directly caused by Ipi-DM1 ADC, we stained B220$^+$ B cells and Foxp3$^+$ Tregs for human CTLA-4 in both human CTLA-4 knock-in (*Ctla4$^{h/h}$*) and WT mice to evaluate expression of CTLA-4 in B cells. As expected, CTLA-4 was detected in Foxp3$^+$ Tregs of *Ctla4$^{h/h}$* but not B cells (*Figure 1—figure supplement 5*).

We then investigated the impact of anti-CTLA-4 ADC in lymphoid organs. Treg percentage and absolute numbers were not impacted by Ipi-DM1 treatment compared to control in the spleen at day 9. The percent of B cells decreased significantly, although the reduction in absolute number of B cells was not statistically significant (*Figure 1—figure supplement 6A and B*). Flow analysis of lymph nodes after Ipi-DM1 treatment resulted in a significant increase in absolute cell numbers of Tregs and B cells and B cell percentage. (*Figure 1—figure supplement 6C and D*).

We then asked how Ipi-DM1 ADC induced Treg impairment impacted B cell lymphopoiesis. Bone marrow cells were stained for flow cytometry and gated on CD45 and T-cell subsets were defined as CD8 (CD8$^+$B220$^-$), CD4 (CD4$^+$B220$^-$), Tregs (CD4$^+$B220$^-$ Foxp3$^+$), and CD4-nonTregs (CD4$^+$B220$^-$ Foxp3$^-$; *Figure 1—figure supplement 2B*). Ipi-DM1 depletion of Tregs was confirmed in the bone marrow (*Figure 1J*). Additionally, total CTLA-4 level in bone marrow Tregs decreased, however Ki67 staining did not change significantly compared to control group (*Figure 1K and L*). In contrast, Ipi-DM1 ADC did not alter bone marrow CD4-nonTreg cell percentage among CD45$^+$, absolute cell number, or CTLA-4 level compared hIgGFc control (*Figure 1—figure supplement 3C and D*). FACS analysis of bone marrow B cells revealed the loss of mature B cells but not immature transitional type1 or Pre-pro/Pro/Pre B cells from Ipi-DM1 treated mice (*Figure 1M–Q*, *Figure 1—figure supplement 7*). Corresponding to mature B cell loss, the Pre-Pro/Pro/Pre B cell subtype percentage in B220 are

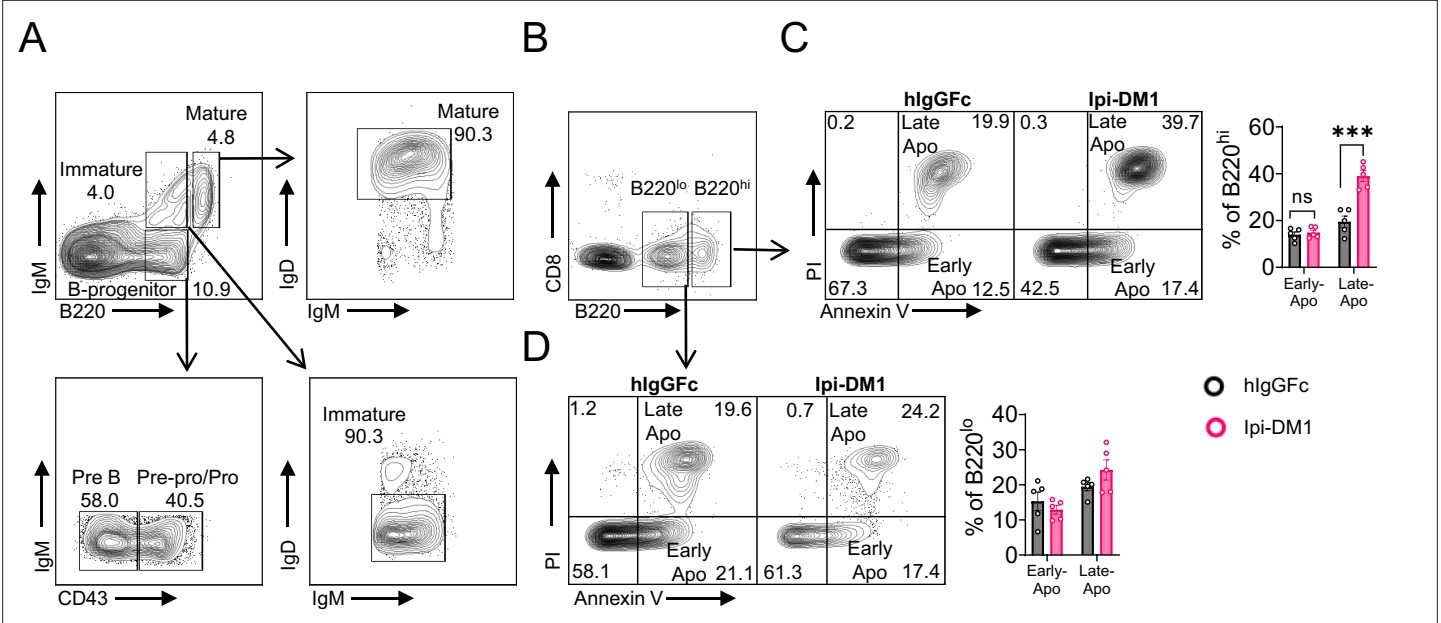

**Figure 2.** CTLA-4 antibody-drug conjugate increases apoptosis in mature B cells in bone marrow. Flow analysis of B cell apoptosis on day 9 after treatment with hIgGFc or Ipi-DM1 ADC. (**A**) FACS profile defining B220$^{hi}$ and B220$^{lo}$ B cell populations in bone marrow (**B**) Gating strategy for B220$^{hi}$ (mature B cells) and B220$^{lo}$ (progenitor/immature B cells) for apoptosis analysis. (**C, D**) FACS profiles and summaries depicting apoptosis of B cells in (**C**) B220$^{hi}$ bone marrow cells, (**D**) B220$^{lo}$ bone marrow cells. Data representative of two independent experiments (n=5) and analyzed using an unpaired two-tailed Student's t test and represented as mean ± SEM. Non-significant (ns) *p<0.05, **p<0.01, ***p<0.001, ****p < 0.0001.

The online version of this article includes the following source data and figure supplement(s) for figure 2:

**Source data 1.**

**Figure supplement 1.** B cell apoptosis in blood and lymphoid organs.

**Figure supplement 1—source data 1.**

enriched (*Figure 1P and Q*, *Figure 1—figure supplement 7*). Additionally, CD21/35 expression level in the remaining mature B cells in Ipi-DM1 treated mice was lower than those from hIgGFc-treated control mice (*Figure 1R*).

We then investigated B cell apoptosis in peripheral blood and lymphoid organs (*Figure 2*, *Figure 2—figure supplement 1*). Only mature bone marrow B220$^{hi}$ B cells showed increase in apoptosis from Ipi-DM1 treatment, while progenitor/immature bone marrow B220$^{lo}$ B cells were similar to control group. Blood and spleen B cell apoptosis were similar to hIgGFC control group, while lymph node B cell apoptosis was lower than the control group. Thus, the loss of B cells in the peripheral blood is likely due to loss of mature B cells in the bone marrow, and that such loss correlates with Treg depletion.

Taken together, data in *Figures 1 and 2* showed that Ipi-DM1 can impair Treg function by depleting Tregs, preferentially the proliferating Tregs with higher CTLA-4 levels, and loss of B cells in the blood and bone marrow while B cell progenitors are not impacted. B cell death increase is observed in only B220$^{hi}$ bone marrow B cells, which are predominantly mature B cells.

## B-cell depletion correlates with Treg impairment by Ipi-DM1 and increases immunoglobulins

To investigate the kinetic relationship between the decline of B cells and Treg impairment, human CTLA-4 knock-in (*Ctla4$^{h/h}$*) mice were treated with control hIgGFc or Ipi-DM1 and bled according to schedule diagram in *Figure 3A*. We observed that peripheral blood samples from mice treated with Ipi-DM1 had a gradual decrease in total leukocytes compared hIgGFc control group followed by a full recovery by Day 25 (*Figure 3B*). The large decrease of CD45 cells in Ipi-DM1 group is predominantly attributed to the decline of B cells, which plateaus on days 9 and 13 and recovers by day 25 (*Figure 3C and D*) in correlation with the kinetics of Tregs (*Figure 3E and F*). Similar to the bone marrow data

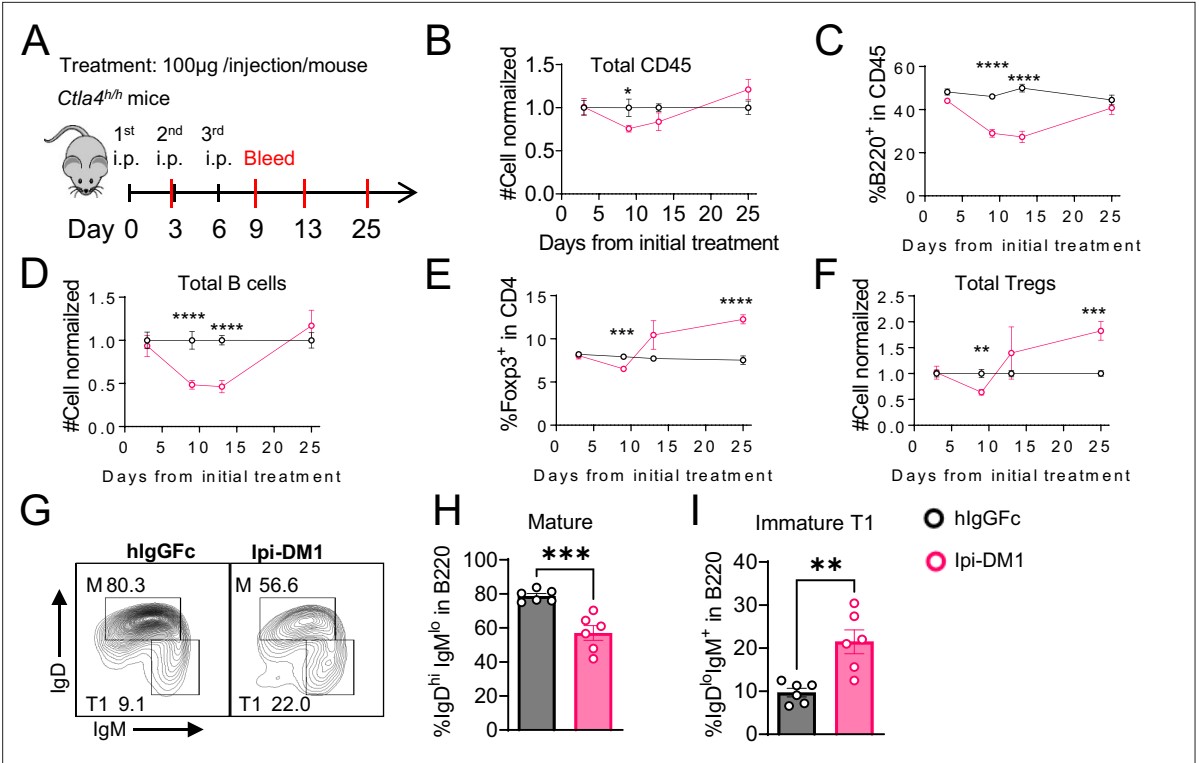

**Figure 3.** B-cell depletion is transient and correlated with a decrease in regulatory T-cell. (**A**) Diagram of experimental design, *Ctla4^h/h* mice were treated intraperitoneal (i.p.) (100 µg/mouse) with hIgGFc or Ipi-DM1 every three days and mice were bled on day 3,9,13, 25 for downstream flow analysis. (**B–F**) Flow data of time course of CD45, B, and Treg cells in peripheral blood, (**B**) CD45 normalized cell number, (**C**) % of B-cells in CD45, (**D**) B-cell normalized cell number, (**E**) % Foxp3+ Tregs in CD4, (**F**) Treg normalized cell number. (**G–J**) Flow analysis of B-cell subtypes from day 13 peripheral blood. (**G**) FACS profile of Mature (**M**) (IgD^hi IgM^low), and immature Transitional type 1 (**T1**) (IgD^low IgM+) B cell subtypes after gating on CD45+B220+ B cells. (**H**) % of mature B cells in B220. (**I**) % of T1 B cells in B220. (**B–F**) Data combined from two independent experiments (n=10). (**G–I**) Data representative of two independent experiments (n=6). Data were analyzed by an unpaired two-tailed Student's t test and represented as mean ± SEM. *p<0.05, **p<0.01, ***p<0.001, ****p < 0.0001.

The online version of this article includes the following source data and figure supplement(s) for figure 3:

**Source data 1.**

**Figure supplement 1.** Treg impairment increases plasma immunoglobulins.

**Figure supplement 1—source data 1.**

**Figure supplement 2.** CTLA-4 antibody-drug conjugate leads to enlarged spleen and lymph nodes.

**Figure supplement 2—source data 1.**

in *Figure 1*, peripheral blood mature B cells decreased and transitional type1 B cells were enriched in Ipi-DM1 group compared hIgGFc control (*Figure 3G–I*). Additionally, Treg impairment by Ipi-DM1 increases immunoglobulins especially IgE (*Figure 3—figure supplement 1*). The lack of Treg depletion in the spleen and lymph nodes at these time points remains to be explained, although significant immune activation, as suggested by splenomegaly and adenopathy (*Figure 3—figure supplement 2*), may result in production of cytokines that drive Treg proliferation.

## Depletion of T-cells but not macrophage rescues B-cells from ablation by Ipi-DM1

We then evaluated how Ipi-DM1-mediated impairment of Treg function affects total CD4 and CD8 T cells in peripheral blood. Mice treated with Ipi-DM1 ADC did not affect the total percentage of CD4 or CD8 T cells among CD45+ leukocytes, but the cell number of both cell types decreased (*Figure 4A and B*). Analysis of lymphoid organ T cells showed differing T cell absolute number phenotype where bone marrow, thymus and spleen T cells were similar to control hIgGFc while lymph node T cells increased (*Figure 4—figure supplement 1*). Staining of Ki67 in CD4 and CD8 revealed a greater

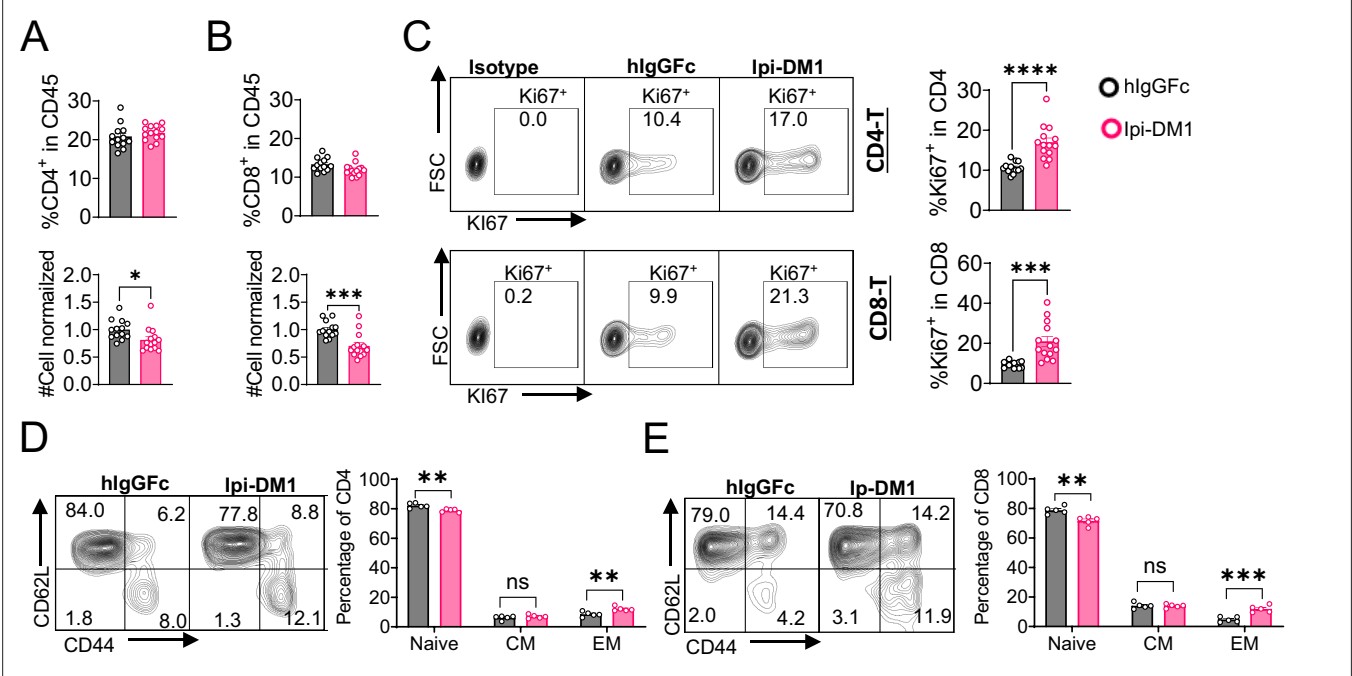

**Figure 4.** CTLA-4 antibody-drug conjugate leads to T-cell activation. Flow data analysis of peripheral blood from *Ctla4$^{h/h}$* mice on day 9 after treatment with hIgGFc or Ipi-DM1 ADC. (**A**) % CD4$^+$ in CD45 and normalized cell number. (**B**) % CD8$^+$ in CD45 and normalized cell number. (**C**) % Ki67$^+$ in CD4 and CD8 T cells. (**D, E**) FACS profiles and summaries depicting the increase in effector T-cells after Ipi-DM1 treatment, naïve (Q1: CD44$^{low}$ CD62L$^{hi}$), central memory (Q2: CD44$^{hi}$ CD62L$^{hi}$) and effector (Q3: CD44$^{hi}$ CD62L$^{low}$), phenotype of CD4$^+$ T cells (**D**), and CD8$^+$ T cells (**E**). (**B–C**) Data combined from two independent experiments (n=13–14). (**D–E**) Data representative of two independent experiments (n=5). Data analyzed using an unpaired two-tailed Student's t test and represented as mean ± SEM. Non-significant [ns], *p<0.05, **p<0.01, ***p<0.001, ****p < 0.0001.

The online version of this article includes the following source data and figure supplement(s) for figure 4:

**Source data 1.**

**Figure supplement 1.** CTLA-4 antibody-drug conjugate impact on T cells in lymphoid organs.

**Figure supplement 1—source data 1.**

**Figure supplement 2.** CTLA-4 antibody-drug conjugate increase T cell proliferation in lymphoid organs.

**Figure supplement 2—source data 1.**

**Figure supplement 3.** T-cell quantity and proliferation are not impacted by IgG-DM1 treatment.

**Figure supplement 3—source data 1.**

percentage of T cells in hyper-proliferative state in blood and lymphoid organs (*Figure 4C*, *Figure 4—figure supplement 2*). Alternatively, mice treated with hIgGFc-DM1 payload control did not change CD4 or CD8 percentage of CD45, cell numbers and Ki67 staining compared to hIgGFc control in the periphery (*Figure 4—figure supplement 3*).

We then analyzed the functional subsets of T cells in mice that received control hIgGFc or Ipi-DM1 at day 9. Using CD44 and CD62L markers, we observed an expansion of effector memory T cells (CD44$^{hi}$CD62L$^{low}$) in Ipi-DM1 group for both CD4 and CD8 T cells (*Figure 4D and E*). Correspondingly, the frequency of naïve T cells was reduced for Ipi-DM1 while central memory T cells were unchanged (*Figure 4D and E*). These results show that CTLA-4 ADC can impair Treg function thereby resulting an increase in effector memory T cells and a hyper-proliferative state similar to animals that lack the expression of CTLA-4 or Foxp3.

Since B cells are devoid of CTLA-4, Ipi-DM1 must have induced other effector cells that are directly responsible for B cell depletion. To understand which cell types are responsible for the destruction of B cells, we depleted either T cells or macrophages using either anti-Thy1.2 mAb or chlondrosome. *Ctla4$^{h/h}$* mice were treated with control hIgGFc, Ipi-DM1, Ipi-DM1 in combination with anti-Thy1.2, or Ipi-DM1 in combination with chlondrosome and bled on day 9 (*Figure 5A*). As shown in the *Figure 5B*, top panel, anti-Thy1.2 mAb efficiently depleted total T cells while macrophage depletion

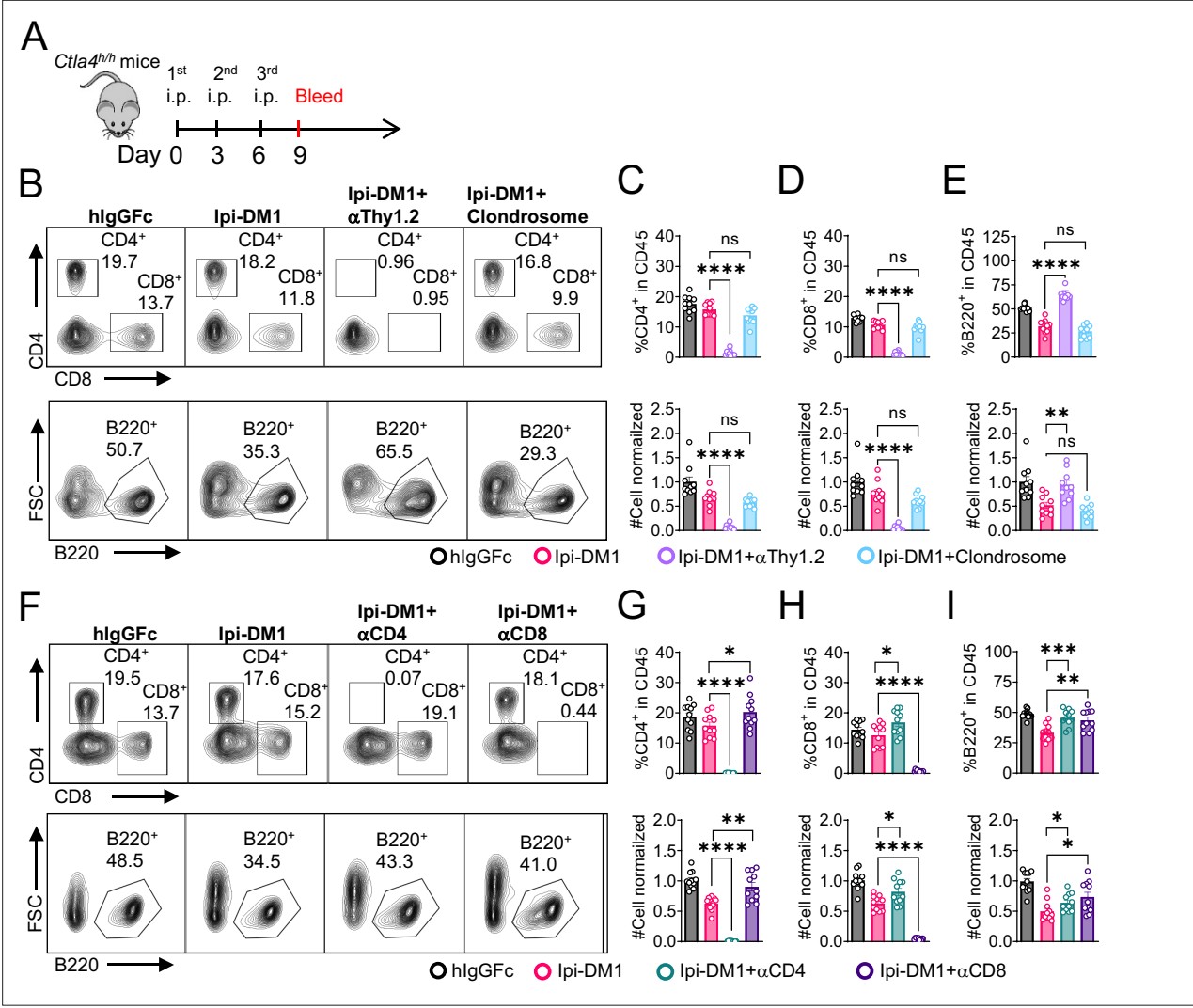

**Figure 5.** B-cell depletion depends on T-cells but not macrophage. (**A**) Diagram of experimental design, male or female *Ctla4^{h/h}* mice were treated intraperitoneal (i.p.) (100 µg/mouse) with hIgGFc or Ipi-DM1 with/out (T-cell depleting antibody 100 µg/mouse of anti-Thy1.2 or 150 µL/mouse of clondrosome or depleting antibody CD4 or CD8 100 µg/mouse) every 3 days and mice were bled on day 9 for downstream flow analysis. (**B**) FACS profiles depicting gating strategy after gating on CD45 for CD8 & CD4 T (top panel) and B cells (bottom panel). (**C–E**) Data summaries from FACS profile(panel B), (**C**) % CD4⁺ in CD45 and normalized cell number, (**D**) % CD8⁺ in CD45 and normalized cell number, and (**E**) % B220⁺ in CD45 and normalized cell number. (**F**) FACS profiles depicting gating strategy after gating on CD45 for CD8 & CD4 T (top panel) and B cells (bottom panel). (**G–I**) Data summaries from FACS profile(panel F), (**G**) % CD4⁺ in CD45 and normalized cell number, (**H**) % CD8⁺ in CD45 and normalized cell number, and (**I**) % B220⁺ in CD45 and normalized cell number. (**B–E**) Data combined from two independent experiments (n=10) and analyzed by ordinary one-way ANOVA with Tukey's multiple comparisons test and represented as mean ± SEM. Non-significant [ns], *p < 0.05, **p < 0.01, ***p < 0.001, ****p < 0.0001. (**F–I**) Data combined from two independent experiments (n=11) and analyzed by unpaired two-tailed Student's t test and represented as mean ± SEM. *p < 0.05, **p < 0.01, ***p < 0.001, ****p < 0.0001.

The online version of this article includes the following source data for figure 5:

**Source data 1.**

by chlondrosome had no effect (*Figure 5B* top panel, 5C, 5D). Additionally, combination of Ipi-DM1 with T cell depleting antibody resulted in a remarkable rescue of B cells as indicated by percentage B220 in CD45 and cell number while combination with macrophage depletion did not rescue the B cells (*Figure 5B* bottom panel, 5E).

In order to understand which subsets of T cells played a critical role in the T cell mediated destruction of B cells, we treated *Ctla4^{h/h}* mice with hIgGFc control, Ipi-DM1, Ipi-DM1 with CD4 depleting antibody, or Ipi-DM1 with CD8 depleting antibody (*Figure 5A*). Both CD4 and CD8 depleting antibodies

provide efficient depletion (*Figure 5F*, top panel) of their respective targeted T cell. Depletion of either CD4 or CD8 T cells results in B cell increase (*Figure 5F* bottom panel, *Figure 5I*). Interestingly, the cell number of CD4 and CD8 T cells increase reciprocally by depletion of the other subsets (*Figure 5G and H*).

Taken together our CTLA-4 antibody-drug conjugate Ipi-DM1 can impair Treg function, which results in a T-cell-mediated destruction of B cells. Whole T cell depletion with anti-Thy1.2, depletion of CD4 or CD8 T cells rescued B cells from abrogation by Ipi-DM1.

## B-cell rescue by Belatacept suggest a role for B7-CD28 interaction in B-cell depletion

The process for T cell activation requires two signals. The first signal is the binding of T-cell receptor (TCR) to antigen-bound major histocompatibility complex (MHC) and the second signal is a costimulatory molecule CD28 binding with B7-1/2 (CD80/86) on the antigen-presenting cell. Having shown that the destruction of B cell can be prevented by total depletion of T cell, we sought to rescue the B cells by breaking the CD28-B7 signal with a soluble CTLA-4 that can bind to B7 but not Ipilimumab or Ipi-DM1 ADC. As shown in *Figure 6—figure supplement 1*, Abatacept can neutralize Ipilimumab/ADC while Belatacept does not. Furthermore, we previously showed that human CTLA4-Ig or mutants can bind to murine B7-1/2 (CD80/86) and block their binding to murine CD28 efficiently (*Du et al., 2018*; *Liu et al., 2023*). Therefore, Belatacept can be used to test if activation of T cells is required for B cell depletion.

We treated mice with Ipi-DM1 to impair Treg function and used Belatacept to break CD28/B7 signal to block T cell activation. Briefly, *Ctla4^{h/h}* mice were treated with hIgGFc or Ipi-DM1 with/out Belatacept and bled on day 9 for flow analysis. Foxp3$^+$ Tregs percentage in CD4 and cell number decreased similarly for Ipi-DM1 with/out Belatacept compared hIgGFc control (*Figure 6A and B*). Additionally, Belatacept did not impact Ipi-DM1 to target CTLA-4 expressing Tregs, as Ipi-DM1 with/out Belactacept had similar total CTLA-4 level reduction compared to control group (*Figure 6C*). However, Belatacept rescued B cells from Ipi-DM1-mediated depletion (*Figure 6D and E*). Correspondingly, Belatacept reduced effector memory T cells (*Figure 6F and G*) as well as granzyme B and IFN-γ expression in CD4 and CD8 T cells compared to Ipi-DM1-treated group (*Figure 6H–K*, *Figure 6—figure supplement 2*).

Collectively, the data presented here show that Ipi-DM1 can impair Treg function resulting in T cell activation as indicated by increase in effector memory T cell markers, GranzymB, and cytokine production thereby resulting in the loss of B cells. However, under Ipi-DM1 induced Treg impairment the addition of mutant soluble CTLA-4-Ig, Belatacept, can reduce T cell activation and thereby preserving B cells.

## B-cell depletion is partially rescued by anti-TNF-alpha

Since T cells are the effector cells responsible for B cell abrogation, it is of great interest to evaluate the molecular mechanism by which B cells are eliminated by T cells. To address this issue, we tested if either FasL or TNF-alpha, which are produced by activated T cells are responsible. To answer this question, *Ctla4^{h/h}* mice were treated with hIgGFc control or Ipi-DM1 with/out Adalimumab a human anti-TNF-alpha that also binds to mouse TNF-alpha (*Figure 7A and B*). Flow analysis showed Tregs were decreased similarly between Ipi-DM1 with/out Adalimumab compared hIgGFc control group (*Figure 7C* top panel, 7D). Remarkably, the anti-TNF-alpha partially rescued B cells as shown by percentage difference of B cell marker B220 between Ipi-DM1 and in combination with Adalimumab as well as cell number (*Figure 7C* bottom panel, 7E). Peripheral blood samples stimulated with Iononmycin/PMA increased intracellular cytokines TNF-alpha and IFN-gamma for T cells from Ipi-DM1 treated mice compared to hIgGFc control, while that with Ipi-DM1 in combination with Adalimumab had slight increase but the change was insignificant except for IFN-gamma in CD8 T cells (*Figure 7F and G*). In contrast, blocking FAS-L with antibody did not result in B cell rescue (*Figure 7—figure supplement 1*).

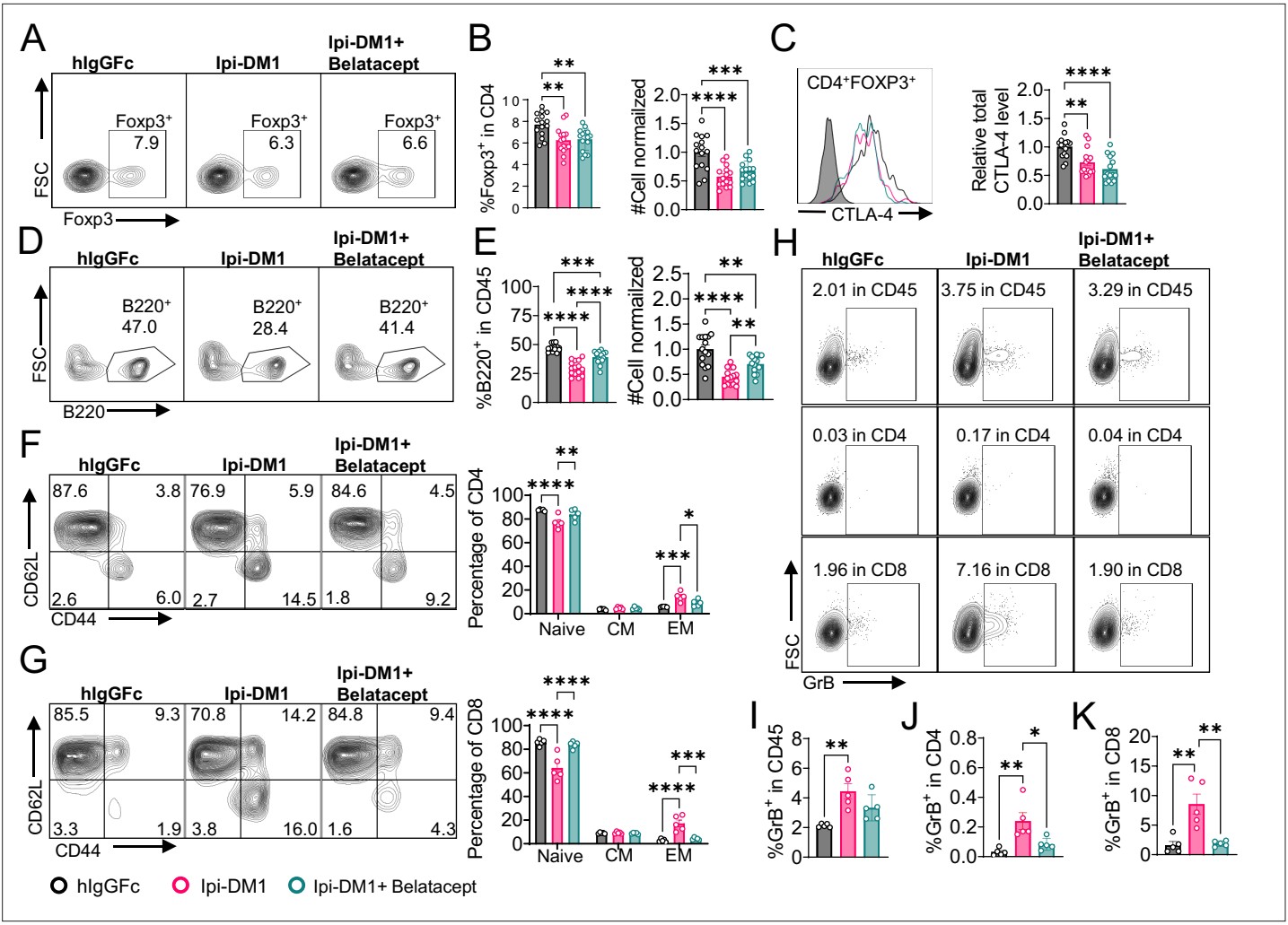

**Figure 6.** Mutant soluble CTLA-4-Ig rescues B-cell. *Ctla4h/h* mice were treated intraperitoneal (i.p.) (100 μg/mouse) with hIgGFc or Ipi-DM1 with/out (100 μg/mouse of Belatacept) every 3 days for total of three doses and mice were bled on day 9 for downstream flow analysis.(**A, D**) FACS profiles depicting gating strategy after gating on CD45 for (**A**) Tregs (% Foxp3+ in CD4 T cells) and (**D**) B cells (% B220+ in CD45). (**B**) % Foxp3+ (Tregs) in CD4 and normalized cell number. (**C**) Representative FACS profile of CTLA-4 expression in Tregs (CD4+ Foxp3+) and Relative CTLA-4 summary. (**E**) % B220+ in CD45 and normalized cell number. (**F–G**) FACS gate and summaries showing mutant CTLA-4-Ig can decrease effector memory T-cells associated with Ipi-DM1 treatment, naïve (Q1: CD44low CD62Lhi), central memory (Q2: CD44hi CD62Lhi) and effector (Q3: CD44hi CD62Llow), phenotype of CD4+ T cells (**F**), and CD8+ T cells (**G**). (**H**) FACS of GranzymeB gating in CD45 (Top panel), CD4 (middle panel), and CD8 (bottom panel). (**I–K**) GranzymeB expression summaries in (**I**) CD45, (**J**) CD4, and (**K**) CD8 cells. (**A–E**) Data combined from three independent experiments (n=15). (**F–K**) Data representative of two independent experiments (n=5). Data analyzed by ordinary one-way ANOVA with Tukey's multiple comparisons test and represented as mean ± SEM. Non-significant [ns], *p < 0.05, **p < 0.01, ***p < 0.001, ****p < 0.0001.

The online version of this article includes the following source data and figure supplement(s) for figure 6:

**Source data 1.**

**Figure supplement 1.** Mutant soluble CTLA-4-Ig does not neutralize Ipilimumab or its drug-conjugate.

**Figure supplement 1—source data 1.**

**Figure supplement 2.** T-cell cytokine production.

**Figure supplement 2—source data 1.**

## Discussion

We have reported that pH-insensitive anti-CTLA-4 antibodies trafficked to, and were degraded in the lysosomes (*Zhang et al., 2019*). Since lysosomal degradation is needed to release DM1 from the antibody for cytotoxicity, the pH-insensitive Ipilimumab was chosen for preparation of antibody-drug conjugate to be used to impact Treg function. We found that in *Ctla4h/h* mice anti-CTLA-4 ADC,

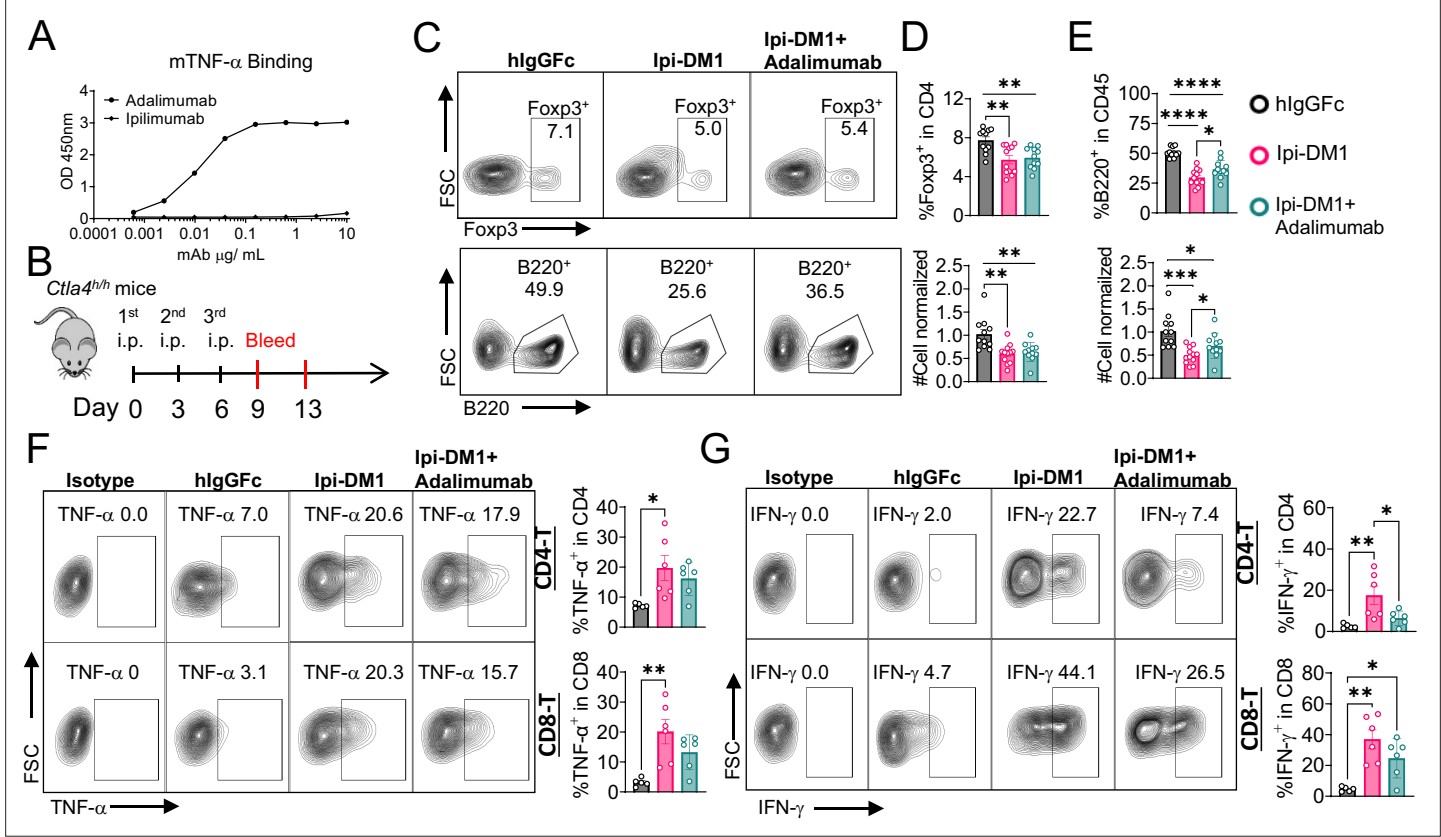

**Figure 7.** B-cell depletion is partially rescued by anti-TNF-alpha. (**A**) ELISA binding of clinical grade drug Adalimumab (Humira) to pre-coated 1 μg/mL mouse-TNF-alpha and detected with anti-hIgG-HRP, Ipilimumab negative control (n=2). (**B**) Diagram of experimental design, male *Ctla4[h/h]* mice were treated intraperitoneal (i.p.) (100 μg/mouse) with hIgGFc or Ipi-DM1 with/out (100 μg/mouse of Adalimumab) every three days for total of three doses and mice were bled on days 9 and 13. (**C**) FACS profiles depicting gating strategy after gating on CD45 for Tregs (% Foxp3[+] in CD4) (top panel) and B cells (% B220[+] in CD45) (bottom panel) from day 9 peripheral blood. (**D**) % Foxp3[+] (Tregs) in CD4 and normalized cell number. (**E**) % B220[+] in CD45 and normalized cell number. (**F, G**) Day 13 peripheral blood post red blood lysis by ACK buffer were cultured and stimulated in the presence of Iononmycin/ PMA, and GolgiPlug for 4 hr followed by intracellular cytokine detection (TNF-alpha, IFN-gamma) in CD4 and CD8 T cells, (**F**) TNF-alpha, and (**G**) IFN-gamma. (**C–E**) Data combined from two independent experiments (n=11–12) and analyzed using an unpaired two-tailed Student's t test and represented as mean ± SEM. *p<0.05, **p<0.01, ***p<0.001, ****p < 0.0001. (**F, G**) Data representative of two independent experiments (n=5–6) and analyzed by ordinary one-way ANOVA with Tukey's multiple comparisons test and represented as mean ± SEM. *p < 0.05, **p < 0.01, ***p < 0.001, ****p < 0.0001.

The online version of this article includes the following source data and figure supplement(s) for figure 7:

**Source data 1.**

**Figure supplement 1.** Anti-FASL fails to rescue B cell.

**Figure supplement 1—source data 1.**

Ipi-DM1, can recapitulate the phenotype of loss of B cells associated clinical deficiency of CTLA-4 (*Kuehn et al., 2014*; *Schubert et al., 2014*)/recycling partner LBRA (*Lo et al., 2015*). Our analysis revealed B cell loss in both bone marrow and peripheral blood, specifically mature B cells. Loss of mature peripheral B cells is consistent with clinical data for some patients with CTLA-4 haploinsufficiency (*Kuehn et al., 2014*; *Schubert et al., 2014*).

B cell loss was previously reported in mice with either naturally occurred (in Scurfy mice) or targeted mutation of Foxp3 and Treg depletion (*Clark et al., 1999*; *Chen et al., 2010*; *Chang et al., 2012*; *Riewaldt et al., 2012*; *Leonardo et al., 2010*; *Pierini et al., 2017*). The kinetics of B cell loss and Treg depletion in the Ipi-DM1 treated mice revealed that B cell loss was transient and correlated to Treg impairment (proliferation, CTLA-4 level, cell number). Our data further showed that B cell destruction under Treg impairment conditions is T-cell-mediated and required T cell activation.

Scurfy mice have poor central and peripheral B lymphopoiesis; however, neonatal WT Treg adoptive transfer in scurfy mice resulted in robust population of mature B cells in the spleen (*Riewaldt*

*et al., 2012*). Genetic ablation of TCRα gene by crossing scurfy with TCRAα$^{-/-}$ mice efficiently support B cell lymphopoiesis and was sufficient to restore B cells in the bone marrow and peripheral compartments (*Riewaldt et al., 2012*). Additionally, reconstitution of bone marrow chimera from scurfy or wild type mixed with that from µMT mice that is deficient in B cells results in normal B cells after 8 weeks from reconstitution (*Chang et al., 2012*). This is in line with our data that Treg impairment results in activated T-cell-mediated apoptosis of mature B cells. Mature B cell loss in bone marrow explains the loss of B cell in the PBL and rules out the possibility that the loss of B cell in the periphery is merely a consequence of defective circulation of B cells normally produced in bone marrow. Others have implied that under Treg depletion, activated T cells are targeting interleukin 7 (IL-7) secreting ICAM1 +perivascular stromal cells needed to progress B cell progenitors (*Pierini et al., 2017*). IL-7 is essential for B lymphopoiesis transition from Pro-B cell into Pre-B cell (*von Freeden-Jeffry et al., 1995*). However, our model system is distinct where only mature B cells are depleted, while B cell progenitor precursors, and specifically Pre-B cells are not impacted. The involvement of T cells is demonstrated by T cell depletion and requirement for T cell activation is demonstrated by requirement for B7-CD28 interaction for B cell loss. Furthermore, we found that anti-TNF-α can partially rescue B-cells from loss imposed by Treg impairment. These data showed that TNF-α is one of the mediators of B cell loss. Additional studies are needed to reveal other mechanism for B cells loss in Treg-defective environment.

While in vitro studies by others showed B cell killing by activated CD25$^+$CD4 Tregs, but not by CD25$^-$ CD4 T cells, *Zhao et al., 2006* this is not the case in vivo where B cell loss is associated with Foxp3 Treg loss or impairment (*Clark et al., 1999*; *Chen et al., 2010*; *Chang et al., 2012*; *Riewaldt et al., 2012*; *Leonardo et al., 2010*; *Pierini et al., 2017*). Furthermore, our current study shows that both CD4 and CD8 T cells contributed to B cell loss. The concept that B cells are actively eliminated by T cells in vivo expands our understanding of T-B cell interaction by showing antagonism rather than just 'immunological help' from T cells to B cells.

## Materials and methods

### Experimental animals
C57BL/6 mice that express the CTLA-4 protein with 100% identity to human CTLA-4 protein under the control of endogenous mouse *Ctla4* locus, *Ctla4$^{h/h}$*, have been previously described (*Lute et al., 2005*). All animals used in experiments were 7–9 weeks age (female mice were used unless male mice are indicated). No blinding or randomization was used and mice were fairly distributed into different treatment groups so that initial average weight of each group were similar. All mice were maintained at the Research Animal Facility of the Institute of Human Virology at the University of Maryland Baltimore School of Medicine. All the animals were handled according to approved institutional animal care and use committee (IACUC) protocol (0221021) of the University of Maryland Baltimore.

### Cell culture and treatment
Wild-type CHO (CHO-WT) cell was previously purchased from ATCC. No cell lines used in this study were listed in the database of cross-contaminated or misidentified cell lines suggested by International Cell Line Authentication Committee (ICLAC). CHO-WT and CHO expressing OFP/human CTLA-4 (CHO-hCTLA-4) were authenticated by visual inspection by microscopy having epithelial cell morphology and CHO-hCTLA-4 was positive for orange fluorescence protein (OFP) while CHO-WT was not. CHO cells tested negative for Mycoplasma by LookOut Mycoplasma PCR Detection Kit (Sigma-Aldrich) according to manufacturer protcol. CHO cells that were stably transfected with human CTLA-4 have been reported (*Du et al., 2018*; *Zhang et al., 2019*). CHO cells were grown in DMEM (Dulbecco's Modified Eagle Medium, Gibco) supplemented with 10% FBS (Hyclone), 100 units/mL of penicillin and 100 µg/mL of streptomycin (Gibco). Mice Peripheral blood leukocytes were cultured in RPMI-1640 medium (containing 10% FBS and 2% penicillin/streptomycin). All cell lines were incubated at 37 °C and were maintained in an atmosphere containing 5% $CO_2$.

### Viability assay
Wild type CHO (CHO-WT) or human CTLA-4 expressing CHO (CHO-hCTLA-4) cells were seeded at 1000 cell/well in a flat 96-well plate at 37 °C for 24 hr in cell culture incubator. The medium was

then replaced with fresh medium containing 1/4 serially diluted vehicle (PBS), Ipilimumab or Ipilimumab-DM1 ADC and cell were incubated at 37 °C for additional 72 hr. Each treatment group was in duplicate or triplicates. CCK-8 viability dye was then added to each well according to manufacturer's protcol and incubated for another (2–2.5 hr) at 37 °C. Wells were subsequently read for absorbance at 450 nM on Spectramax ID3 Molecular Devices plate reader. In case of testing whether soluble human CTLA-4-Ig or mutant can neutralize Ipilimumab or Ipilimumab-DM1, the same conditions above were used except that Abetacept or Belatacept concentrations were kept constant at 6 μg/mL. Data is normalized according to the following equation ($treatment_{OD450}$ - $background_{averageOD450}$ /$vehicle_{averageOD450}$ - $background_{average}$) x 100.

## Cell surface CTLA-4 binding

Freshly trypsinized human CTLA-4 expressing CHO (CHO-hCTLA-4) cells were stained with 1/4 serially dilute anti-CTLA-4 Iplimiumab or Iplimiumab-DM1 ADC in FACS buffer (2% FBS with 2 mM EDTA) for 30 min on ice. Cells were then washed twice with FACS buffer and incubated with anti-human IgG AF488 secondary antibody for 20 min on ice. Cells were washed twice with FACS buffer and processed on BD Canto II flow cytometer. In the case of whether soluble human CTLA-4-Ig or mutant can neutralize Ipilimumab or Ipilimumab-DM1 and prevent them from binding cell to surface CTLA-4 the same conditions above were used except that Abetacept or Belatacept concentrations were kept constant at 6 μg/mL.

## Peripheral blood T cell stimulation

Peripheral blood samples 50 μL were treated with ACK Lysis buffer and washed with RPMI-1640 medium. Leukocytes were then stimulated with 1 μg/ml each of phorbol 12-myristate 13-acetate (PMA) (Sigma-Aldrich, St. Louis, MO), ionomycin (Sigma Aldrich, St. Louis, MO) and BD GolgiStop (BD Biosciences, cat. 51-2092KZ) and cultured in RPMI-1640 medium (containing 10% FBS and 2% penicillin/streptomycin) for 4 hr at 37 °C in 96-well plate. Medium was removed and cells were washed twice with FACS buffer (2% FBS with 2 mM EDTA) followed by surface staining and fix/perm and intracellular staining.

## Flow Cytometry

Leukocytes from blood or bone marrow were FACS stained directly or after red blood cell lysis with ACK buffer. Fc Receptor was blocked with anti-FCR clone 2.4G2 at 10 μg/mL in FACS buffer for 10 min at room temperature and respective surface staining antibodies cocktails were added to each sample and incubated on ice for additional 20 min. Cells were then washed twice with 1xPBS and stained with 1 x live/Dead Fixable dye Aqua in 1 x PBS for 7 min at room temperature. Cells were then washed twice with FACS buffer and fixed with eBioscience Foxp3/Transcription Factor Staining Buffer Set for 40 min. Samples were either washed twice and resuspended in FACS buffer and processed for flow acquisition or further permeabilized for intracellular staining using perm buffer from same kit (eBioscience Foxp3/Transcription Factor Staining Buffer Set). In General, all intracellular staining of Foxp3, Ki67, hCTLA-4, GranzymeB, or cytokines were done overnight at 4 degree. In the case of detecting intracellular cytokines, samples were cultured and simulated according to the above protocol followed by surface then intracellular cytokine staining. Apoptosis staining with Annexin V/Pi was performed according manufacturer protocol after surface staining. Samples were acquired by the BD Canto II Flow cytometer and data were analyzed by Flowjo software.

## ELISAs

96-well high-binding polystyrene plates were pre-coated with 50 μL of 1 μg/mL of His-hCTLA-4, or mouse TNF-alpha in coating buffer (0.1 M bicarbonate) at 4 °C overnight. After washing away the unbound protein/antibody thrice with 0.05% PBST, the plates were blocked with blocking buffer (1% BSA in PBST) for 1 hr at room temperature. All primary antibody incubation was done in blocking buffer at 4 °C overnight or room temperature for 1 hr. For coated His-hCTLA-4 a given concentration of either Ipilimumab/DM1, or hIgGFc/DM1; For coated mouse TNF-alpha a given concentration of Adalimumab or Ipilimumab negative control were used. Following primary incubation plates were then washed with PBST for four times and incubated with a goat anti-human IgGFc HRP conjugate secondary antibody at 1/20,000 dilution for detection in blocking buffer for 1 hr at room temperature.

Plates were then washed 4 times with PBST followed by development with 1-Step Ultra TMB-ELISA Substrate Solution for 10 min and stopped with 2 N sulfuric acid. For Immunoglobulin quantification, an ELISA kit was used following manufacturer protocol. Wells were read at 450 nM on Spectramax ID3 Molecular Devices plate reader.

## Antibody-drug conjugate preparation

Ipilimumab or hIgGFc were buffer exchanged using prepacked column PD-10 into 1 X PBS. Ipilimumab or hIgGFc 5 mg each at concentration 1.5 mg/mL were conjugated with SMCC-DM1(15 eq) in 1 x PBS in the presence of 10% DMSO at 37 °C under mild shaking conditions for 50 minutes to furnish Ipilimumab-DM1 or hIgGFc-DM1 ADC respectively. Reaction mixture was cleaned up from excess SMCC-DM1 by buffer exchange with a in house ADC buffer at pH 6.5 containing (20 mM Histidine, 8% sucrose, and Polysorbate 80 0.02%) by Econo-Pac 10DG prepacked column according to manufacturer protocol. Confirmed fraction containing ADC by nano drop were combined and concentrated down to 1 mL using Pierce Protein Concentrator PES, (30 K for hIgGFc-DM1) or (50 K for Ipilimumab-DM1) MWCO according manufacture protocol followed by sterile filtration. ADC concentration was determined using BCA assay according to manufacturer protocol. ADC were diluted to 0.4–0.5 mg/mL in 1xPBS and $A_{280}$ &$A_{252}$ were recorded on Nanodrop One. The drug to antibody ratio (DAR) was calculated following previous literature (*Chen, 2013*).

## Antibodies and fusion proteins used for in vivo studies

CTLA-4-Ig fusion proteins were synthesized by Sydlabs, Inc (Boston, MA). Recombinant Ipilimumab with amino acid sequence disclosed in WC500109302 was produced by Sydlabs Inc (Boston, MA). Azide-free human IgG-Fc was purchased from Athens Research and Technology (Athens, GA, USA). Antibody-drug conjugates Ipilimumab-DM1 and hIgGFc-DM1 were prepared from parent antibodies in lab. Depletion antibodies anti-mouse Thy1.2, clone 30H12(BE0066); anti-mouse CD4, clone GK1.5 (BE0003-1); and anti-mouse CD8α, clone 2.43 (BE00-61) were purchased by Bioxcell Inc (West Lebanon, NH, USA). Anti-mouse FASL, clone MFL3 (BE00319) was purchased by Bioxcell Inc (West Lebanon, NH, USA). Anti-TNF-α, Adalimumab, clinical grade Humira was purchased from Premium Health Services (Columbia, MD, USA).

## Other reagents and material

SMCC-DM1 drug payload with cross linker (Cederlanelabs/Cayman Chemical Co, 23926–10) Anti-mouse FcγR, clone 2.4G2 (Bioxcell Inc, BE0307). Clodrosome (Encapsula Nano Sciences, SKU# CLD-8909). Mouse anti-human IgGFc secondary antibody (Invitrogen/Thermo Fisher Scientific, 05-42-00). Goat anti-human IgGFc (HRP) preadsorbed (Abcam, ab98624). Polyhistidine-tagged human CTLA-4 (HIS-hCTLA-4) (Sino Biological Inc, 11159-H08H). Mouse TNF-alpha (Sino Biological Inc, 50349-MNAE). IgE ELISA Kit (Thermo Fisher Scientific/Invitrogen, 88-50460-86). IgG ELISA Kit (Thermo Fisher Scientific/Invitrogen, 88-50400-86). IgM ELISA Kit (Thermo Fisher Scientific/Invitrogen, 88-50470-86). IgA ELISA Kit (Thermo Fisher Scientific/Invitrogen, 88-50450-86). 123 eBeads counting beads flow (Thermo Fisher Scientific/Invitrogen, 01-1234-42). NuPage BIS-TRIS gels 4–12% (Thermo Fisher Scientific/Invitrogen, NP0335BOX). NuPAGE MOPS SDS Running Buffer (20 X) (Thermo Fisher Scientific/Invitrogen, NP0001). NuPage LDS sample buffer (4 x) (Thermo Fisher Scientific/Invitrogen, NP0007). Protein Ladder (Thermo Fisher Scientific, BP3603500). Sucrose (Sigma-Aldrich, S0389-1KG). L-Histidine (Sigma-Aldrich, H8000-10G). Dimethyl sulfoxide (DMSO) (Santa Cruz Biotechnology, Sc-258801). Polysorbate 80 (Fisher Scientific, L13315). Buffer exchange prepacked column PD-10 (GE health care/ Cytiva Life Sciences, 17085101). Buffer exchange prepacked column Econo-Pac 10DG (Bio-Rad, 7322010). Cell Counting Kit-8 (CCK-8) (Bimake, B34304). 1-Step Ultra TMB-ELISA Substrate Solution (Thermo Fisher Scientific, 34028). Gibco ACK Lysing buffer (Thermo Fisher scientific/Gibco, A1049201). Live/Dead Fixable Aqua Dead Cell Stain (Thermo Fisher Scientific/Life Technologies, L34966). eBioscience Foxp3 /Transcription Factor Staining Buffer Set (Thermo Fisher Scientific/Invitrogen, 00-5523-00). Pierce Protein Concentrator PES, 30 K MWCO, 2–6 mL (Thermo Fisher Scientific/Pierce, 88521) Pierce Protein Concentrators PES, 50 K MWCO, 2–6 mL (Thermo Fisher Scientific/Pierce, 88538). Pierce BCA Protein Assay Kit (Thermo Fisher Scientific/Pierce, 23227). LookOut Mycoplasma PCR Detection Kit (Sigma-Aldrich, MP0035).

## Flow antibodies

eBioscience/ Thermo Fisher Scientific (San Diego, CA): APC-eFlour780 anti-mouse CD45, clone 30-F11 (47-0451-82);eFlour450 anti-mouse CD4, clone GK1.5 (48-0041-82); Pacific Blue anti-mouse CD4, clone RM-4–4 (116008); PE-Cyanine7 anti-mouse CD8α, clone 53–6.7 (25-0081-82); PE-Cyanine7 anti-mouse CD8β, clone H35-17.2 (25-0083-82);PerCP-Cy5.5 anti-mouse B220, clone RA3-6B2 (45-0452-82); APC anti-mouse Foxp3, clone FJK-16s (17-5773-82);FITC anti-mouse Ki67, clone SolA15 (11-5698-82);FITC Isotype rat IgG2ak, clone eBR2a (11-4321-82);FITC anti- mouse CD44, clone IM7 (11-0441-85);PerCP-Cy5.5 anti-mouse CD62L, clone MEL-14 (45-0621-82);PE-Cyanine7 anti-mouse B220, clone RA3-6B2 (25-0452-82); APC anti-mouse IgM, clone II/41 (17-5790-82); eFlour 450 anti-mouse CD21/CD35, clone 4E3 (48-0212-82); AF488 anti-mouse TNF-α, clone MP6-XT22 (53-7321-82); AF488 Isotype rat IgG1k, clone eBRG1 (53-4301-80); APC anti-mouse IFNγ, clone XMG1.2 (17-7311-82);APC Isotype rat IgG1k, clone eBRG1 (17-4301-82); FITC anti- mouse GranzymeB, clone NGZB (11-8898-82); Alexa Fluor 488-conjugated goat anti-human IgG (H+L) cross-adsorbed secondary antibody (A-11013). BioLegend (San Diego, CA): PE anti-human CTLA-4, clone BNI3 (369604); PE Isotype mIgG2ak, clone MPC-173 (400212); PerCP-Cy5.5 anti-mouse IgD, clone 11–26 c.21 (405710); BV421 anti-mouse CD21/CD35, clone 7E9 (12342); APC Annexin V Apoptosis Detection Kit with PI (640932).

## Statistical analysis

The specific tests used to analyze each set of experiments are indicated in the figure legends. All samples are biological replicates. Data were analyzed using a two-tailed t-test to compare between two groups, one-way or two way ANOVA (analysis of variance) for multiple comparisons. In the graphs, y-axis error bars represent S.E.M. or S.D. as indicated. Statistical calculations were performed using GraphPad Prism software (GraphPad Software, San Diego, California). Non-significant [ns], *$p<0.05$, **$p<0.01$, ***$p<0.001$, ****$p<0.0001$. Reported CTLA-4 level and absolute cell numbers are normalized by the average value of hIgGFc control group.

# Acknowledgements

This study is supported by grants from the National Institutes of Health (R01AI154722) and OncoC4, Inc .

# Additional information

### Competing interests

Wei Wu: Employee of OncoC4, Inc. Pan Zheng, Yang Liu: Co-founder of OncoC4, Inc. The other authors declare that no competing interests exist.

### Funding

| Funder | Grant reference number | Author |
| --- | --- | --- |
| National Institutes of Health | R01AI154722 | Lishan Su |
| OncoC4, Inc | | Mingyue Liu |

The funders had no role in study design, data collection and interpretation, or the decision to submit the work for publication.

### Author contributions

Musleh M Muthana, Conceptualization, Data curation, Formal analysis, Supervision, Validation, Investigation, Visualization, Methodology, Writing – original draft, Writing – review and editing; Xuexiang Du, Methodology, Writing – review and editing, Harvested blood/tissue samples; Mingyue Liu, Methodology, Writing – review and editing, Harvested blood/tissue samples; Xu Wang, Methodology, Writing – review and editing, Harvested blood/tissue samples; Wei Wu, Methodology, Writing – review and editing, harvest blood/tissue samples; Chunxia Ai, Methodology, Writing – review and editing,

Harvested blood/tissue samples; Lishan Su, Supervision, Funding acquisition, Writing – review and editing; Pan Zheng, Conceptualization, Supervision, Writing – review and editing; Yang Liu, Conceptualization, Supervision, Funding acquisition, Writing – original draft, Writing – review and editing

### Author ORCIDs
Musleh M Muthana  https://orcid.org/0000-0003-1242-5631

### Ethics
This study was performed in strict accordance with the recommendations in the Guide for the Care and Use of Laboratory Animals of the National Institutes of Health. All of the animals were handled according to approved institutional animal care and use committee (IACUC) protocol (0221021) of the University of Maryland Baltimore.

Reviewer #1 (Public Review): https://doi.org/10.7554/eLife.87281.3.sa1
Reviewer #2 (Public Review): https://doi.org/10.7554/eLife.87281.3.sa2
Reviewer #3 (Public Review): https://doi.org/10.7554/eLife.87281.3.sa3
Author Response https://doi.org/10.7554/eLife.87281.3.sa4

## Additional files

### Supplementary files
• MDAR checklist
• Supplementary file 1. Table S1. Drug to antibody ratio (DAR).

### Data availability
Source data contain the numerical data used to generate the figures.

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
