## [Editor Report · eLife assessment]

This **valuable** study presents presents **solid** evidence that an anti-CTLA-4 antibody drug conjugate transiently depletes circulating B cells in a mouse model, showing how dysregulation of the T cell immune system can impact B cell homeostasis. The work will be of broad interest to immunologists and medical biologists, but a major limitation is that the mechanism of B-cell reduction remains unclear, as evidence of killing of B-cells by T-cells is not presented.

---

## [Referee Report · Reviewer #1 (Public Review)]

The manuscript by Muthana et al. describes the effect of injection of an antibody specific for human CTLA4 conjugated to a cytotoxic molecule (Ipi-DM1) in knock-in mice expressing human CTLA4. The authors show that Ipi-DM1 administration causes a partial decrease (about 50% in absolute number) of mature B cells in blood and bone marrow 9-14 days after the beginning of treatment. B cell progenitors and pre-B cells in the BM are not affected. Ipi-DM1 also results in a partial decrease in Foxp3+ Tregs (about 40% in absolute number) and a slight increase in activation of conventional T cells (Tconvs) in the blood, spleen, BM and LNs at D9 as well as increased plasma immunoglobulins especially IgE. Tconv depletion, CTLA4-Ig or anti-TNF mAb partially prevents the effect of ipi-DM1 on B cells. This effect of Ipi-DM1 on the reduction B cells and Tregs at D9 is not observed in the spleen and lymph nodes (maybe not the good timing to see it), and there is even an increase in the number of Treg and the frequency and number of B cells in lymph nodes. This work is interesting but has the following major limitations:

1- This work could have been of more interest if the Ipi-DM1 molecule would be used in the clinic. As this is not the case, the intimate mechanism of the effect of this molecule in mice is of reduced interest.

2- The fact that a partial deletion of Tregs is associated with activation of Tconvs and a decrease in B cells is not new. According to the authors, their work would be the first to show that activation of Tconvs would lead to B cell death. However, this is shown in an indirect way and the mechanisms are not really elucidated. The experiments to try to show a causal link are of 2 types: deletion of T cells (Fig 5) and blocking T cell activation with CTLA4-Ig (Fig 6). These 2 experiments are not fully convincing. The absence of B cell depletion in the blood when T cells are deleted can be explained by other mechanisms, such as B cell recirculation to lymphoid tissues or an effect of massive T cell death for example. The experiment with CTLA4-Ig is more convincing because the effect is targeted to activated T cells only. However, the prevention of B cell ablation is only partial. Since only blood is analyzed, other mechanisms could explain the B cell loss, such as their recirculation in lymphoid tissues.

3- The authors propose that the drop in B cell numbers in the blood in mice treated with Ipi-DM1 results from reduced mature B cells in the bone marrow. However, B cells are continuously recirculating between the blood and secondary lymphoid tissues. The drop of blood B cells could be well explained by an increased recirculation to lymphoid organs. The increased numbers of B cells in lymph nodes support this latter hypothesis.

4- The new Figure 2 suggests direct evidence of apoptosis of mature B cells in the BM of treated mice using a PI/annexin V staining assay. This is an important point to support the point of the manuscript. However, using the same assay, the level of B cell apoptosis is of 80% in lymph nodes and 50% in the spleen in control mice (see new Figure 2-figure supplement 1), which is way too high and questions the reliability of this assay. It is likely that B cells enter apoptosis only in vitro due to some artefactual stress.

---

## [Referee Report · Reviewer #2 (Public Review)]

Despite the fact that CTLA-4 is a critical molecule for inhibiting the immune response, surprisingly individuals with heterozygous CTLA-4 mutations exhibit immunodeficiency, presenting with antibody deficiency secondary to B cell loss. Why the loss of a molecule that regulates T cell activation should lead to B cell loss has remained unclear. In this study, Muthana and colleagues use an anti-CTLA-4 antibody drug conjugate (aCTLA-4 ADC) to delete cells expressing high levels of CTLA-4, and show that this leads to a reduction in B cells. The aCTLA-4 ADC is found to delete a subset of Tregs, leading to hyperactivation of T cells that is associated with B cell depletion. Using blocking antibodies, the authors implicate TNFa in the observed B cell loss.

The reciprocal regulation of T and B cell homeostasis is an important research area. While it has been shown that Treg defects are associated with B cell loss, the mechanisms at play are incompletely understood. CTLA-4 is not normally expressed in B cells so an indirect mechanism of action is assumed. The authors show that the decrease in Treg following aCTLA-4 ADC treatment is associated with activation of T cells, and that B cell loss is blunted if T cells are depleted. A role for both CD4 and CD8 T cells is identified by selective CD4/CD8 depletion. T cells appear to require CD28 costimulation in order to mediate B cell loss, since the response is partially inhibited in the presence of the costimulation blockade drug belatacept (CTLA-4-Ig). Finally, experiments using the anti-TNFa antibody adalimumab suggest a potential role for TNFa in the depletion of B cells.

While the manuscript makes a useful contribution, a number of limitations remain. Perhaps most important is the extent to which this model mimics the natural situation in individuals with CTLA-4 mutations (or following CTLA-4-based clinical interventions). aCTLA-4 ADC treatment permits acute deletion of Treg expressing high levels of CTLA-4, whereas in patients the Treg population remains but is specifically impaired in CTLA-4 function. Secondly, although the requirement for T cells to mediate B cell loss is convincingly demonstrated, the incomplete reversal by TNFa blockade suggests additional unidentified factors contribute to this effect. Finally, although the manuscript favours peripheral killing of mature B cells over alterations to B cell lymphopoiesis, one concern is that this may simply reflect the model employed: the short-term (6 day) treatment used here may be too acute to alter B cell development, but this may nevertheless be a feature of prolonged immune dysregulation in humans.

---

## [Referee Report · Reviewer #3 (Public Review)]

The co-suppressive molecule CTLA-4 has a critical role in the maintenance of peripheral tolerance, primarily by Treg mediated control of the co-stimulatory molecules CD80 and CD86. As stated by the authors, previous studies have found a variety of effects of anti-CTLA-4 antibody treatment or genetic loss of CTLA-4 on B-cells. These include increased B-cell activation and antibody production, autoantibody production, impairment of B-cell production in the bone marrow and loss of peripheral B-cells. In this article Muthana et al use a CTLA-4 humanized mouse model and examine the effects of drug conjugated CTLA-4 on the immune system. They observe a transient loss of B-cells in the blood of the treated mice. They then use a range of immune interventions such as T-cell depletion and blocking antibodies to demonstrate that this effect is dependent on T-cell activation.

Since anti-CTLA-4 immunotherapy is in active clinical use exploration of its effects are welcome, this is helped by the use of a humanized CTLA-4 system which should be considered a strength of the paper. However, currently the central premise of this paper, that B-cells are depleted seems underexplored. Direct evidence of T-cell killing of B-cells is never presented, rather it is inferred from the reduced numbers of B-cells in the blood and increased apoptosis in the bone marrow. It is not made clear if B-cell numbers in the bone marrow are reduced.

Upon examining lymphoid organs it seems that the spleen is relatively unchanged while the lymph nodes have a large increase in B-cells alongside increased serum antibody levels. The paper does underline the importance of looking at the differences of multiple immune compartments and interesting phenomenon are described in each compartment. Simultaneous inhibition of B-cell lymphopoiesis and blood trafficking with strong activation and antibody production of lymphoid resident (presumably germinal center) B-cells appears to be occurring. However the current overall interpretation that B-cells are broadly depleted is perhaps too simplistic and largely ignores the lymphoid organs and serum antibodies.

---

## [Author Response]

The following is the authors’ response to the original reviews.

**Reviewer #1 (Public Review):**
The manuscript by Muthana et al. describes the effect of injection of an antibody specific for human CTLA4 conjugated to a cytotoxic molecule (Ipi-DM1) in knock-in mice expressing human CTLA4. The authors show that Ipi-DM1 administration causes a partial decrease (about 50% in absolute number) of mature B cells in blood and bone marrow 9-14 days after the beginning of treatment. Ipi-DM1 also results in a partial decrease in Foxp3+ Tregs (about 40% in absolute number) and a slight increase in activation of conventional T cells (Tconvs) in the blood at D9. Tconv depletion, CTLA4-Ig or anti-TNF mAb partially prevents the effect of ipi-DM1 on B cells. This work is interesting but has the following major limitations:1. This work could have been of more interest if the Ipi-DM1 molecule would be used in the clinic. As this is not the case, the intimate mechanism of the effect of this molecule in mice is of reduced interest.

The goal of the current study is to use Ipi-DM1 ADC as probe to study mechanism of B cell loss observed in Treg-deficient host.

1. The fact that a partial deletion of Tregs is associated with activation of Tconvs and a decrease in B cells has been published several times and is therefore not new. According to the authors, their work would be the first to show that activation of Tconvs would lead to B cell depletion. However, this is shown in an indirect way and the mechanisms are not really elucidated. Indeed, this work shows a correlation between an increase in Tconv activation and a decrease in the number of B cells in the blood. The experiments to try to show a causal link are of 2 types: deletion of T cells (Fig 4) and blocking T cell activation with CTLA4-Ig (Fig 5) (neutralization of TNF addresses another question). Neither of these 2 experiments is totally convincing. Indeed, the absence of B cell depletion when T cells are deleted can be explained by other mechanisms than the preservation of B cell destruction by activated T cells. The phenomenon could be explained by B cell recirculation to lymphoid tissues or an effect of massive T cell death for example. The experiment shown in Fig. 5 with Belatacept is more convincing because this time the effect is targeted to activated T cells only. However, the prevention of B cell ablation is only partial. Again, since only blood is analyzed, other mechanisms could explain the B cell loss, such as their recirculation in lymphoid tissues.

While the concept of treg depletion leads to activation of Tconv cells and reduced B cells has been previously published, B cell loss was explained on basis of defective B cell lymphopoiesis due to low production of stroma cell-derived IL-7 or destruction of stromal cells by effector T cells. Our new data established that loss of B cells in the context of Treg depletion was not due to defects in the number of pre-/pro-B cells. Rather it is the death of mature B cells in the bone marrow.

To address the reviewer’s concern that the B cell loss was merely caused by a change in circulating pattern, we performed a new study on the effect of the ADC on B cells in bone marrow. Our new data reveal loss of mature bone marrow B cells, and that such loss is associated with increased apoptosis of mature B cells. Therefore, the loss of B cells in the peripheral blood is not due to a changed circulation.Furthermore, our data show that B cell progenitor, Pre-B, cells are not changed. Therefore, B cell lymphopoiesis is not the reason for B cell loss in our model system.

1. It is disappointing that only the blood (and sometimes the bone marrow) was studied in this work. The interest of doing experiments in mice is to have access to many tissues such as the spleen, lymph nodes, colon, lung, and liver. To conclude that there is B cell deletion without showing lymphoid organs (where the majority of B cells reside) is insufficient. As discussed above, the drop in B cells in the blood could be due to their recirculation in lymphoid organs. In addition, there is no measurement of functional B cells activity. Do mice treated with Ipi-DM1 have a decreased ability to develop an antibody response following immunization?

We have analyzed lymph nodes and spleen at the same time points. Unfortunately, Treg depletion was no longer observed at these time points. As expected, we did not see a clear depletion of B cells (Figure 1-figure supplement 6). In regards to functional B cell activity, we observed an increase of plasma immunoglobulins especially IgE which are now shown in Figure 3-figure supplement 1.

1. Although it is difficult to study in vivo, there is not a single evidence of increased B cell death after injection of Ipi-DM1.

Figure 2 & Figure 2-supplement 1 provides B cell death comparisons between IpiDM1 and hIgGFc group for bone marrow, blood, spleen, and lymph nodes. Statistically significant increase in B cell death is observed in mature B cells in bone marrow.

1. In most of the experiments, B cells are quantified with the B220 marker alone, but this marker, in some cases, can be expressed by other cells. It would have been preferable to use a marker more specific to B cells such as CD19 for example.

We have added data to support the death of mature B cells using other markers.

Minor points.1. It should be indicated whether human CTLA4 binds normally to mouse CD80 CD86. We do not know if knock-in mice with human CTLA4 have a fully functional immune system.

We have indicated this point as suggested and cited our previous work line 226-227 (ref 23 & 24)

1. The manuscript is too long. Some of the data in the figures should be moved to supplemental figures. This is the case, for example, for some trivial stainings (Fig 1F, Fig 4B, 4F, Fig 5A, D, F, G). The figure legends and the Materials and Methods section are far too long. On the other hand, Fig 5-Fig Sup 1 could go into the main figures.

The figure legends, materials, and methods may be too long, but our intention is to provide as much info as possible for others who may be interested in our model system.

1. The anti-CTLA4 ADC reagent should be better explained and defined in the text.

The anti-CTLA-4 ADC reagent synthesis described in materials/methods under“Antibody-drug conjugate preparation.”

**Reviewer #2 (Public Review):**
Despite the fact that CTLA-4 is a critical molecule for inhibiting the immune response, surprisingly individuals with heterozygous CTLA-4 mutations exhibit immunodeficiency, presenting with antibody deficiency secondary to B cell loss. Why the loss of a molecule that regulates T cell activation should lead to B cell loss has remained unclear. In this study, Muthana and colleagues use an anti-CTLA-4 antibody drug conjugate (aCTLA-4 ADC) to delete cells expressing high levels of CTLA-4, and show that this leads to a reduction in B cells. The aCTLA-4 ADC is found to delete a subset of Tregs, leading to hyperactivation of T cells that is associated with B cell depletion. Using blocking antibodies, the authors implicate TNFa in the observed B cell loss.The reciprocal regulation of T and B cell homeostasis is an important research area. While it has been shown that Treg defects are associated with B cell loss, the mechanisms at play are incompletely understood. CTLA-4 is not normally expressed in B cells so an indirect mechanism of action is assumed. The authors show that the decrease in Treg following aCTLA-4 ADC treatment is associated with activation of T cells, and that B cell loss is blunted if T cells are depleted. A role for both CD4 and CD8 T cells is identified by selective CD4/CD8 depletion. T cells appear to require CD28 costimulation in order to mediate B cell loss, since the response is partially inhibited in the presence of the costimulation blockade drug belatacept (CTLA-4-Ig). Finally, experiments using the anti-TNFa antibody adalimumab suggest a potential role for TNFa in the depletion of B cells.While the manuscript makes a useful contribution, a number of questions remain. Perhaps most important is the extent to which this model mimics the natural situation in individuals with CTLA-4 mutations (or following CTLA-4-based clinical interventions). aCTLA-4 ADC treatment permits acute deletion of Treg expressing high levels of CTLA-4, whereas in patients the Treg population remains but is specifically impaired in CTLA-4 function. Secondly, although the requirement for T cells to mediate B cell loss is convincingly demonstrated, the incomplete reversal by TNFa blockade suggests additional unidentified factors contribute to this effect. Finally, although the manuscript favours peripheral killing of mature B cells over alterations to B cell lymphopoiesis, one concern is that this may simply reflect the model employed: the shortterm (6 day) treatment used here may be too acute to alter B cell development, but this may nevertheless be a feature of prolonged immune dysregulation in humans.

We appreciate reviewer’s comments and the difference between short-term depletion and permanent inactivation of Treg by genetic mutation is discussed. We would note that apart from mutation, dynamic Treg perturbation does occur under autoimmune conditions. Therefore, our data have significant implications for T-B cell interactions.

TNF-alpha is implicated in B cell loss as evidenced by the partial rescue with Anti-TNF treatment. We did not try to exclude the possibility that other mechanisms are involved.

Our data shows loss of circulating B cell in peripheral blood and mature bone marrow B cells. B cell progenitor, Pre-B, cells are not changed due Ipi-DM1 induced treg impairment, therefore Bcell lymphopoiesis is not the reason for B cell loss in our model system. Evidence of increased cell death is only observed in mature B cells (Figure 2).

1. Following aCTLA-4 ADC treatment, it is surprising how subtle the deletion of Treg is (from ~8% to ~7%, Fig 1G), compared to the marked deletion of CTLA-4-expressing CHO cells. Is this a feature of in vivo versus in vitro treatment? If Treg are treated in vitro is deletion more efficient? How does the expression level of CTLA-4 in the CHO cells compare with the Treg in these assays?

We appreciate reviewer’s comments. The anti-CTLA-4 ADC targets CTLA-4 on cell surface. On average about 5% of Tregs express surface CTLA-4 at given moment while human CTLA-4 expressing CHO cell line stains > 90%. Nevertheless, Treg cell number in peripheral blood is reduced by >40%. Additionally, we have included bone marrow data, which shows a greater percentage of Treg depletion (Figure 1J).

1. The decrease in CTLA-4 seen after ipi-DM1 is complicated by the fact that the control DM1 conjugate (IgG1-DM1) appears to significantly increase CTLA-4 expression (Fig 1 supplement 2). It would be useful to clarify when hIgGFc is used versus hIgGFc-DM1 given the additional complexity introduced here (comparisons lacking a payload differ in more than one variable, while the hIgGFc-DM1 is clearly not inert).

We appreciate reviewer’s comments. We agree that the hIgGFc-DM1 control slightly increased CTLA-4 level; nevertheless, it did not alter B cells, T cells or their proliferation capacity when compared to hIgGFc. Our point here is that B cell depletion is not mediated by DM1 payload off target release (new-version Figure 1-Figure supplement 4, old version Figure 1-figure supplement 2). As for the clarification comment when hIgGFc is used versus hIgGFcDM1 is used, the information is clarified in the figure legend. Comparisons are made between (hIgGFc VS Ipi-DM1) or (hIgGFc VS hIgGFc-DM1).

1. T cell-derived IFNg is another potential contender for influencing B cell homeostasis - have you considered testing whether this also contributes in your model?

We appreciate reviewer’s suggestion. IFNγ was reported to induce apoptosis and cell arrest in Pre- B cells, however these studies are invitro studies Garvey et.al Immunology. 1994 Mar; 81(3): 381–388; Grawunder et.al Eur. J. Immunol. 23, 544–551. Since we did not observe any effect on Pre-B cells, we have not followed the literature to investigate the role of IFNy in B cell loss in our model.

**Reviewer #3 (Public Review):**
The co-suppressive molecule CTLA-4 has a critical role in the maintenance of peripheral tolerance, primarily by Treg mediated control of the co-stimulatory molecules CD80 and CD86. As stated by the authors, previous studies have found a variety of effects of anti-CTLA-4 antibody treatment or genetic loss of CTLA-4 on B-cells. These include increased B-cell activation and antibody production, autoantibody production, impairment of B-cell production in the bone marrow and loss of peripheral B-cells. In this article Muthana et al use a CTLA-4 humanized mouse model and examine the effects of drug conjugated CTLA-4 on the immune system. They observe a transient loss of B-cells in the blood of the treated mice. They then use a range of immune interventions such as T-cell depletion and blocking antibodies to demonstrate that this effect is dependent on T-cell activation.

Since anti-CTLA-4 immunotherapy is in active clinical use exploration of its effects are welcome, this is helped by the use of a humanized CTLA-4 system which should be considered a strength of the paper. However, currently, the central premise of this paper, that B-cells are depleted, seems underexplored. Direct evidence of T-cell killing of B-cells is never presented, rather it is inferred from the reduced numbers of B-cells in the blood. The status of B-cells in sites that contain a large proportion of B-cells such as the spleen and lymph nodes is not examined. Additionally, no examination of B-cell antibody production is performed.

We appreciate reviewer’s comments. To address the reviewer’s concerns we performed additional experiments to evaluate the impact on B cells in other organs, as detailed in our responses to specific questions.

1. Examination of B-cell apoptosis/cell death and T-cell mediated cytotoxicity is needed. The authors repeatedly refer to auto destructive T-cells without ever demonstrating their presence or any direct evidence that B-cells are dying. This is particularly important in the context of the blood since an alternative hypothesis would be a change in B cell trafficking and infiltration into tissues.

We appreciate reviewer’s comments. To address the reviewer’s concern that B cell loss in blood might be caused by a change in B cell trafficking pattern. We performed new study on the effect of the ADC on B cells in bone marrow. Our new data reveal loss of mature bone marrow B cells, and that such loss is associated with increased apoptosis of mature B cells (Figure 2). Therefore, the loss of B cells in the peripheral blood is not due to B cell trafficking and infiltration into tissues.

1. The authors demonstrate that B-cells are mostly reduced in blood at around days 10 to 15, I believe it is critical to determine if this is also reflected in the lymphoid organs such as the spleen and lymph nodes.

We appreciate reviewer’s comments. We have analyzed lymph node and spleen at the same time points. Unfortunately, Treg depletion was no longer observed at these time points. As expected, we did not see a clear depletion of B cells (Figure 1-figure supplement 6).

1. Related to the above point do the authors see evidence of Splenomegaly or lymphadenopathy?

We appreciate reviewer’s comment. Evidence of splenomegaly and lymphadenopathy is presented in Figure 3-figure supplement 2.

1. Minimal examination of the status of the B-cells or antibody production is performed. Previous reports would suggest that plasma cell induction and antibody responses may be expected. Do serum antibody levels change in this system?

We appreciate reviewer’s comment. Increases of plasma immunoglobulins especially IgE are now shown in Figure 3-figure supplement 1.

1. Its unclear how the authors interpret their experiment with anti-TNFa (figure 6). Are they suggesting that TNFa itself depletes B-cells or that it is part of the inflammatory milieu that contributes to wider T-cell activation and, in turn, B-cell depletion?

We have discussed these possibilities in the revised manuscript.